# AutoProtocol: Agent-Driven Automation of CT and MRI Protocoling in Radiology

**Philip Wesp**[*][1]                                                        PWESP@STANFORD.EDU
**Louisa Fay**[*][1]                                                            LFAY@STANFORD.EDU
**Faidra Patsatzi**[1]                                                       FAIDRAP@STANFORD.EDU
**Sergios Gatidis**[1]                                                      SGATIDIS@STANFORD.EDU
[1] *Stanford Center for Artificial Intelligence in Medicine and Imaging (AIMI), Stanford, CA, USA*

## Abstract

In radiology, imaging exam protocoling is a high-volume, expert-driven task that increasingly limits clinical throughput as imaging demand grows. We introduce AutoProtocol, an agentic pipeline that frames protocol generation as a training-free, retrieval-augmented problem. Given a patient's history, the system retrieves similar prior cases from large-scale clinical data and provides them as context to an LLM-based agent, enabling patient-specific protocol generation without task-specific fine-tuning. Using a dataset of 4.4M exams, AutoProtocol achieves strong performance on 500 evaluation cases, reaching 0.84 mean reciprocal rank (MRR) and 76.1% Hits@1 under LLM-based evaluation, and 0.81 MRR / 73.6% Hits@1 using embedding-based similarity (vs. 0.10 MRR / 10% Hits@1 baseline).

**Keywords:** Imaging Exam Protocoling, In-Context Learning, LLM-agents, Radiology

## 1. Introduction

Imaging exam protocoling is a high-stakes, knowledge-intensive step that has become a throughput bottleneck as clinical imaging volumes continue to rise (Kalra et al., 2020; Langlotz, 2025). Each order requires synthesizing unstructured clinical indications with longitudinal patient history and evolving institutional guidelines. Suboptimal protocol choices reduce diagnostic yield, drive repeat imaging, and create backlogs of awaited orders that delay downstream workflows. In previous studies, protocol automation has largely been pursued as a classification problem over a fixed protocol vocabulary, an approach that performs well on routine orders but generalizes poorly to complex cases where the right protocol depends on individual history and analogy to similar patients (Chung et al., 2024). A natural alternative is in-context learning (ICL) with large language models (LLMs), where retrieved patient history and similar prior cases are provided as context, enabling protocol recommendations without task-specific fine-tuning (Brown et al., 2020; Lewis et al., 2021).

We present AutoProtocol, an agentic pipeline (Bluethgen et al., 2025) that implements retrieval-augmented ICL over an institutional database of 4.4M previously acquired clinical imaging exams (Fig. 1). Our contributions are: (i) We formulate imaging protocol generation as a training-free, retrieval-augmented ICL problem requiring no labeled data or task-specific fine-tuning. (ii) We introduce a dual evaluation framework that pairs semantic quality assessment via LLM-judge preference ranking with an embedding-based similarity ranking metric. (iii) We evaluate our pipeline on 500 real CT and MR cases, leveraging the 4.4M-case institutional database for context retrieval.

---

[*] Contributed equally

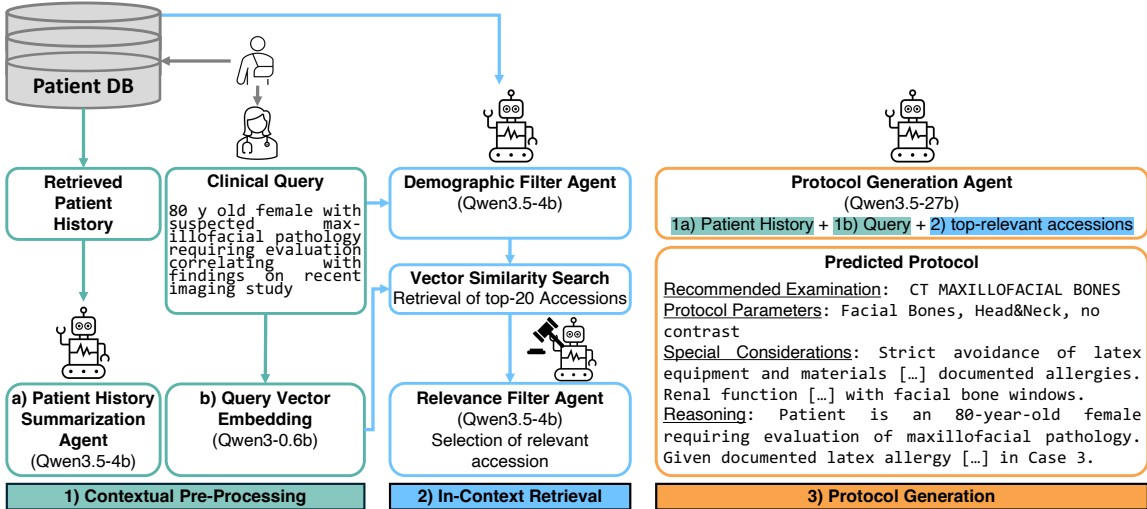

Figure 1: AutoProtocol. A training-free agentic pipeline for radiology protocol generation incorporating (1) contextual pre-processing of patient history and clinical queries, (2) in-context retrieval of relevant previous clinical cases, and (3) LLM-based protocol generation that synthesizes retrieved clinical data into protocol predictions, including reasoning.

## 2. Methods

We introduce AutoProtocol, a three-stage, training-free agentic workflow for radiology protocol generation (Fig. 1). **(1) Contextual Pre-processing.** The input is a free-text clinical query. If available, longitudinal patient records are retrieved from the institutional database and summarized into a clinical narrative using Qwen3-4B (Zhang et al., 2025). Simultaneously, the input query is embedded (Qwen3-Embedding-0.6B). **(2) In-Context Retrieval.** Demographic attributes (age range and sex) are extracted from the query via structured LLM parsing and used to constrain the retrieval candidate pool. The embedded query is then matched against six clinical text fields using cosine similarity, and the top-20 candidates are retrieved. A *Relevance Filter Agent* (Qwen3-4B) refines this set by selecting the most clinically relevant examples. **(3) Protocol Generation.** A single structured LLM call (Qwen3.5-27B), conditioned on the clinical query, patient history, and retrieved similar cases, produces the final structured protocol recommendation including recommended examination, protocol parameters, modality, anatomical region, contrast agent, special considerations, and clinical reasoning.

**Data.** Our pipeline is based on data from a large US hospital System. The dataset comprises 4.4M exams, including 1.5M CT (20.2%) and MR studies (14.4%), and 374,218 patients (female: 53.4%, male: 46.6%, age at examination: 53.3 ± 22.8 years).

**Experiments and Evaluation.** We assess performance using two complementary evaluation strategies. **(1) LLM-judge.** For each query, the judge, an independent LLM (Qwen3.5-9B), receives the ground-truth reference protocol as a reference and 10 candidate predictions, the actual predicted protocol and 9 predictions from other cases of the evaluation dataset. The LLM-based judge ranks all 10 candidates by clinical relevance to the

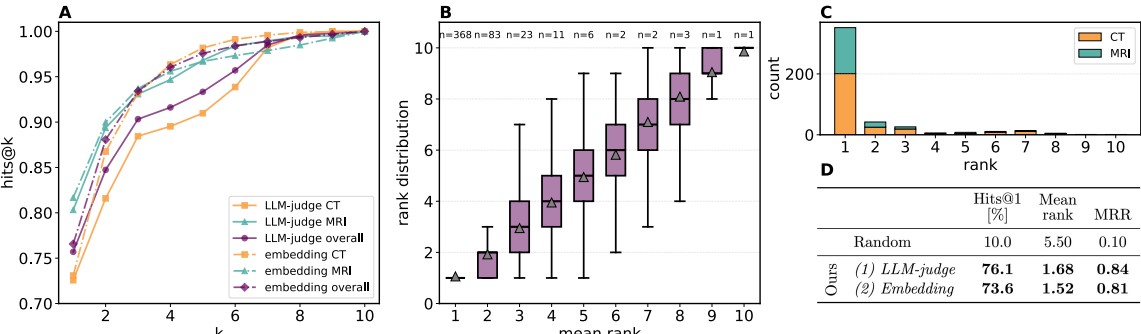

Figure 2: **(A)** Hits@$k$ curves for LLM-judge and cosine similarity evaluation (CT, MRI, overall), compared to the random baseline. **(B)** Distribution of per-sampling embedding ranks grouped by the predicted protocol mean rank across 100 repeated samplings. **(C)** LLM-judge rank distribution, stratified by modality (CT, MRI). **(D)** Hits@1, mean rank, and MRR for the random baseline, LLM-judge, and cosine similarity evaluation.

reference protocol. **(2) Embedding-based similarity ranking.** For each test sample, we embed the predicted protocol and a set of 10 reference protocols (the ground truth and 9 randomly sampled distractors) into a shared embedding space. We compute cosine similarity between the prediction and each reference and rank the ground-truth protocol among the candidates. This process is repeated over 100 samplings to assess stability.

We evaluate our pipeline based on 500 clinical studies, and report, for both approaches, Hits@$k$ ($k = 1$–$10$), mean reciprocal rank (MRR), and mean rank against a random baseline.

## 3. Results

Both evaluation strategies place the correct protocol at the top of the candidate list across test 500 cases (excluded from retrieval), achieving mean ranks of 1.68 (LLM-judge) and 1.52 (embedding-based), with Hits@1 of 76.1% (MRR 0.84) and 73.6% (MRR 0.81), respectively, and Hits@3 exceeding 90% in both settings (Fig. 2D). The strong agreement between LLM-based and embedding-based evaluations suggests that the model produces semantically meaningful protocol recommendations rather than relying on superficial template matching. Modality-wise, MRI consistently outperforms CT under both metrics (Fig. 2A). The embedding-based rank distribution over 100 independent samplings (Fig. 2B) shows low variance, with 368/500 cases consistently ranking the correct protocol at position 1.

## 4. Conclusion

We have demonstrated that image protocoling can be effectively formulated as a retrieval-augmented ICL problem. Leveraging 4.4M real-world exams available in the institutional health record database, AutoProtocol performs retrieval-conditioned inference over patient history and relevant prior cases, enabling accurate protocol recommendations without task- or site-specific fine-tuning. Future work includes retrieval of guidelines, validation on clinical queries, multi-institution generalization, and benchmarking against clinical radiologists.

## Acknowledgments

This work was financially supported by GE HealthCare. Philipp Wesp is funded by the Deutsche Forschungsgemeinschaft (DFG, German Research Foundation) – 553239084.

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
