# OpenReview forum: "AutoProtocol: Agent-Driven Automation of CT and MRI Protocoling in Radiology"
_MIDL.io/2026/Short_Papers — MIDL 2026 - Short Papers Poster_

### Official Review · Reviewer_nVMo · 2026-05-03
**Interesting preliminary results pending a human-reader study**

**Rating:** 5
**Confidence:** 3

**Review:**

The paper is concise and well written, and does not make any unwarranted claims about the work and results.

**Summary:**

This paper proposes and evaluated a label-free pipeline for (text) protocol generation for radiology studies. Utilizing the internal database (millions of records) of the hospital where the work was performed, the author demonstrate that it is possible to generate semantically (evaluated by cosine similarity between embeddings, and LLM sorting) useful and meaningful protocols.

This is a very interesting first step, assuming it would be in the future evaluated by human-readers to assess the clinical relevance and usefulness of the protocols produced.

**Strengths:**

- Real-world clinical problem addressed
- Clever leverage of existing historical data from hospitals, avoiding costly (and unnecessary) new labels production
- Results evaluated with two different metrics

**Weaknesses:**

- No validation and testing of the produced text by human-readers
- The llm production evaluated by another llm could bias the results ; or at least hide potential issues, assuming that both models have the same bias/failure mode

**Justification Of Rating:**

The results and data efficiency are very appealing, but this task and application not being my field, I keep a lower confidence score in my review.

---

### Decision · Program_Chairs · 2026-05-08

Accept (Poster)